# Renal Handling of Albumin—From Early Findings to Current Concepts

**DOI:** 10.3390/ijms22115809

**Published:** 2021-05-28

**Authors:** Jakub Gburek, Bogusława Konopska, Krzysztof Gołąb

**Affiliations:** Department of Pharmaceutical Biochemistry, Wroclaw Medical University, Borowska 211A, 50-556 Wrocław, Poland; jakub.gburek@umed.wroc.pl (J.G.); boguslawa.konopska@umed.wroc.pl (B.K.)

**Keywords:** albumin, renal catabolism of proteins, renal proximal tubule, proteinuria, megalin, cubilin

## Abstract

Albumin is the main protein of blood plasma, lymph, cerebrospinal and interstitial fluid. The protein participates in a variety of important biological functions, such as maintenance of proper colloidal osmotic pressure, transport of important metabolites and antioxidant action. Synthesis of albumin takes place mainly in the liver, and its catabolism occurs mostly in vascular endothelium of muscle, skin and liver, as well as in the kidney tubular epithelium. Long-lasting investigation in this area has delineated the principal route of its catabolism involving glomerular filtration, tubular endocytic uptake via the multiligand scavenger receptor tandem—megalin and cubilin-amnionless complex, as well as lysosomal degradation to amino acids. However, the research of the last few decades indicates that also additional mechanisms may operate in this process to some extent. Direct uptake of albumin in glomerular podocytes via receptor for crystallizable region of immunoglobulins (neonatal FC receptor) was demonstrated. Additionally, luminal recycling of short peptides into the bloodstream and/or back into tubular lumen or transcytosis of whole molecules was suggested. The article discusses the molecular aspects of these processes and presents the major findings and controversies arising in the light of the research concerning the last decade. Their better characterization is essential for further research into pathophysiology of proteinuric renal failure and development of effective therapeutic strategies.

## 1. Introduction

Albumin is one of the earliest recognized proteins in the body. It was first described by Denis in 1840. Its name derives from the Latin word albus (white) due to the property of white precipitate formation in an acidic environment. This protein is very common in the body, and is present in inter alia: plasma, lymph, cerebrospinal and extracellular fluid. It plays many important physiological functions. Albumin accounts for over 50% of all plasma proteins, and its relatively high concentration (45 g/L, 0.6 mM) determines about 80% of the colloid-osmotic pressure of blood plasma [1]. It is responsible for the distribution of water between plasma and the rest of the extracellular fluid, which ensures the maintenance of normal blood hemodynamics and prevents swelling. The transporting function of albumin is equally important. The large volume of distribution associated with its relatively easy penetration through the capillary epithelium, causes that over 60% of the total pool of this protein is present in the extravascular space. Such a large penetration of albumin into interstitial fluids allows its contact with most of the body’s cells, making it an ideal carrier of low-molecular metabolites. Endogenous substances transported by albumin include, inter alia: long-chain fatty acids, aromatic carboxylic acids, bilirubin, bile acids, porphyrins, nitric oxide and ions of divalent metals, including Co, Cu, Ni, Zn cations. Albumin is also a protein which binds the cations of transition metals involved in free radical generation, such as Cu or Fe. It is also characterized by its antioxidative role, and enables complexation of Cd, Hg, and V ions, which is important for detoxification processes. Due to the presence of the free -SH Cys34 group, albumin per se is an antioxidant and important element of plasma antioxidative barrier [2]. It is also worth mentioning that during long-term starvation albumin is degraded to provide essential amino acids for macromolecular synthesis and energy production. In the state of protein deficiency, the synthesis of albumin is 2.5-fold accelerated, and its resources are reduced by almost 50% [3,4].

Taking into consideration the fact that albumin plays significant role in homeostasis, it is essential for the organism to maintain its normal plasma concentration. A decrease in albumin concentration is observed in many metabolic disorders of various etiologies. It may be the result of: improper distribution for example at pancreatitis, reduced synthesis in liver cirrhosis, amino acid absorption disorders or low-protein diets; loss in the course of enteropathy, burns, surgical procedures or nephrotic syndrome, as well as accelerated catabolism in inflammation and cancer. The aforementioned decline is manifested by many serious disorders affecting central nervous system, respiratory system and cardiovascular system, including the formation of edema, activation of coagulation factors, hypotransferrinemia, high-density lipoprotein (HDL) dyslipidemia, oxidative stress and many metabolic disorders associated with its transport function. In addition, in hypoalbuminemic states, the free pharmacologically active fraction of drugs is increased, which should be considered for setting up the optimal dosage. Hyperalbuminemia is a rare condition and can be caused by significant dehydration or excessive venous blood stasis. However, excessive plasma albumin levels are not associated with more serious disorders [5,6].

Intravenous administration of albumin preparations in the state of hypoalbuminemia causes a rapid, temporary correction of the oncotic pressure and prevents hypovolemia. Therefore, albumin preparations have been used, inter alia, in the treatment of extensive burns, acute respiratory failure, a severe hemolytic syndrome in newborns and in patients after cardiac surgery [7,8]. In addition, due to its ability to accumulate in the tissues of tumors and inflammatory sites, the studies on drugs based on conjugates containing albumin were conducted. For example, drugs coupled with exo- or endogenous albumin, cross-linked in the form of albumin micro- and nanocapsules, as well as genetic fusions in the case of polypeptide drugs, were examined [9].

Therefore, albumin catabolism constitutes a significant research problem in terms of pathophysiology and interventional medicine. One of the main organs involved in this process is a kidney. Renal catabolism disorders can lead not only to complications associated with hypoalbuminemia, but also to the development of nephrotic syndrome in the course of albuminuria, and consequently to the terminal failure of this organ. The relation between proteinuria and renal failure is complex and involves series of pathological events mediating by chemokines for interstitial inflammation, mononuclear cell accumulation, and intrarenal activation of complement. Proinflammatory and profibrogenic signals lead to local injuries and related interstitial renal fibrosis. As a consequence, the depletion of functional nephrons is observed [10]. For this reason, this issue has been the subject of extensive research for many years. Studies that have been carried out over the last decade have significantly contributed to the understanding of the molecular mechanisms involved in this process, as well as some controversies associated with them. The study presents the results of the latest research in this field.

## 2. General Albumin Metabolism

### 2.1. Synthesis

Albumin is produced on polysomes of the rough endoplasmic reticulum of hepatocytes and secreted as a preprotein. When moving to the smooth endoplasmic reticulum, a signal peptide is removed. Further processing occurs on the secretory pathway and involves removal of hexapeptide which is present at the N-terminus of the molecule [11]. The rate of albumin synthesis is 10–15 g per day, which is about 10% of the total protein synthesis in the liver.

A small amount of albumin (about 2 g) is stored in the liver, while the majority is secreted into the vascular space. The plasma pool of this protein constitutes 30–40% of its total amount, and the remaining part is found mainly in the skin and muscles. About 5% of albumin leaks into the extracellular space, from which it returns to the systemic circulation by the lymphatic route [4]. The synthesis of albumin is a continuous process which is regulated at the level of transcription and initiation of translation by different stimuli. For example, the synthesis is intensified after food intake and decreases in inter-meal periods. This process is also affected by hormones. The synthesis of albumin increases in hyperthyroidism and decreases in hypothyroidism. Corticosteroids and insulin enhance the production of albumin in healthy people, while the synthesis of this protein is inhibited in acute phase response. A decrease in potassium levels in hepatocytes reduces the amount of albumin released into circulation but does not inhibit the protein synthesis itself. However, the changes in oncotic pressure exhibit the dominant effect on the intensity of albumin synthesis. The correct albumin concentration is also maintained due to the balanced catabolism found in all tissues [12].

### 2.2. Catabolism

The albumin’s half-life in plasma is 19 days, and its daily breakdown in the human body does not exceed 14 g. The catabolism of this protein occurs mainly in the muscles and skin (about 40–60%), and more specifically in the vascular endothelial cells of these tissues. In light of recent studies, it is believed that the molecular mechanism of internalization and transport of albumin to the degradation organelles in these cells is caveolin-dependent endocytosis involving scavenger receptors gp18, gp30 and gp60 (albondin) [13,14,15,16]. The liver is involved in this process to a lesser extent (approximately 15%). In addition to vascular endothelial cells, hepatocytes also participate in albumin uptake. Its catabolism in parenchymal cells also occurs through structures associated with caveolin [17,18]. Proteolytic degradation of albumin takes place after an internalization and fusion of caveolae with lysosomes. Molecules are broken down into free amino acids that feed the systemic pool of amino acids.

There is some evidence that inflammation-associated hypoalbuminemia is rather caused by increased turnover than by diminished synthesis. Oxidized albumin or albumin modified in other ways is broken down in the liver. In inflammatory states, these processes are upregulated also due to increased capillary permeability and intensive flux of plasma protein to interstitium [19].

Albumin catabolism also occurs in kidneys (approximately 10%), however, molecular mechanisms involved in the renal albumin turnover are essentially different. After glomerular filtration, the protein is internalized in the proximal tubules via clathrin-dependent endocytosis with the participation of the tandem of macromolecular scavenger receptors—megalin and cubilin. Then the albumin molecules undergo lysosomal degradation and/or transcytosis, which will be discussed in detail in the following chapters.

## 3. Renal Albumin Catabolism

Renal albumin catabolism involves filtration in glomerulus, reabsorption in proximal tubules and intracellular degradation or partial transcytosis into the bloodstream. Only a small amount of albumin, i.e., up to about 100 mg per day, is excreted in urine. In the case of damage to the filtration barrier or dysfunction of proximal tubules, albumin appears in the urine in concentrations which may exceed 3 g per day [20].

### 3.1. Glomerular Filtration of Albumin

The permeability of the glomerular barrier determines the composition of primary filtrate.

It is formed from three layers: an innermost fenestrated endothelial cell layer, glomerular basement membrane, and an outermost layer of podocytes with their interdigitating foot processes bridged by a slit diaphragm. The multilayer characteristics of the sieve cause the pore to gradually decrease in size. In recent years, thanks to the development of genetic animal models, it has been possible to identify the main components responsible for this barrier integrity. The most important role is currently attributed to some structural components of the glomerular basement membrane (GBM) (e.g., collagen type IV and laminin β2) and podocyte glycocalyx proteins (e.g., podocin and nephrin) [21,22]. However, the function of podocytes is not limited to sieving. It has been demonstrated that both human and animal podocytes endocytose albumin in vitro [23,24]. Data also suggests that podocytes clear proteins from the GBM via transcytosis [25]. These studies suggest that podocytes internalize albumin and other plasma proteins (e.g., IgG) and in this way prevent the glomerular filtration barrier (GFB) from clogging.

The measure of glomerular permeability is the glomerular sieving coefficient (GSC), which is the ratio of concentration of a given molecule in the primary filtrate to its plasma concentration. The degree of albumin permeation is the subject giving rise to many controversies, and the GSC values determined in various experimental models differ by up to three orders of magnitude.

Theoretical calculations show that the pore diameter of this barrier is about 4 nm and is negatively charged due to the relatively high content of glycosaminoglycans. Therefore, the filtration of larger diameter particles and/or those with average negative net charge should be difficult. In terms of shape, the albumin molecule resembles an ellipsoid with large and small diameters of 14 nm and 3.8 nm, respectively, and its resultant charge is −15 (pI = 4.5). In the case of a molecule with the aforementioned characteristics, GSC should be relatively low and range between (5·10^−4^–7·10^−4^) [26,27]. The concentration of albumin in the plasma is about 45 g/L, hence its concentration in the primary filtrate should vary between 22 and 32 mg/L [28]. These values are similar to those obtained in observations in patients with rare diseases affecting albumin renal handling or using animal experimental models. The results obtained using the micro-puncture method of the early proximal tubule sections in healthy rats (6·10^−4^) are highly in line with the theoretical model [27,29]. The GSC of albumin calculated from its urinary concentration from patients with Fanconi syndrome characterized by tubular albuminuria is slightly lower than the lower range of theoretical values (8.0·10^−5^) [30]. In contrast, GSC calculated on the basis of urinary albumin concentration in rats with pharmacologically inhibited tubular reabsorption is slightly higher (3.3·10^−4^) [31]. Moreover, in studies on an isolated kidney perfused at 8 °C, where tubular activity is completely inhibited due to the lack of fluidity of the cell membrane, GSC values were even higher (1·10^−3^) [32]. The aforementioned discrepancies can be explained by technical limitations of the applied measurement methods. For example, with the use of micro-puncture technology, there is a risk that the collected sample may be contaminated with plasma from capillaries damaged during pipette projection, and therefore it does not reflect the actual filtrate due to the very rapid absorption of proteins in the first section of the proximal tubule. In the case of data from patients with Fanconi syndrome, it is not possible to accurately assess the degree of proximal tubule damage and it should be assumed that the reabsorption of proteins is not completely inhibited. The same applies to the models with pharmacological inhibition of reabsorption. Also, data from experiments using isolated organs are difficult to interpret due to completely different hemodynamics than in vivo. Despite discrepancies in GSC values determined by different techniques, the generally accepted paradigm for many years was that glomerular filtration of albumin is relatively small and is characterized by a GSC index below 1·10^−3^.

The results obtained using a low-invasive technique of in situ two-photon microscopy have raised great controversy in recent years. The determined GSC was significantly higher compared to the previous studies (2–4·10^−2^). Fluorescent-labeled albumin was administered intravenously to diabetic *Munich-Wistar* rats with surface glomeruli and normoalbuminemia. GSC was determined by comparing plasma fluorescence intensity in glomerular capillaries and primary filtrate in Bowman space. Similar values were obtained irrespective of the concentration and method of labeled albumin administration (bolus/infusion) [33]. These data are consistent with the previous studies in rats, in which proteinuria was induced by administration of puromycin aminonucleoside. This drug induced hypoalbuminemia manifesting a more than 60% decrease in serum albumin concentration. Compared to healthy subjects, there was no albumin reuptake in the brush border of proximal tubule cells in rats exposed to puromycin, which resulted in an increase in the renal albumin excretion index to above 300 mg per day [34]. The lack of reabsorption was associated with a pronounced decrease in the expression of megalin, V-ATPase and clathrin in the apical pole of the tubule [35]. The authors of the above mentioned studies believe that glomerular albumin filtration is a very efficient process, which is usually based on the principle of stationary convective flow rather than diffusion, and that the limitations resulting from the size of the albumin molecule and structural features of the glomerular barrier are not as large as expected. As a consequence, they suggest that albuminuria is only caused by abnormal albumin reabsorption and is not a result of the damage to the glomerular barrier as previously thought. The alternative model has been presented in a couple of reviews [36,37,38].

The selectivity of the glomerular barrier, which was associated with the charge, was not actually observed in the previous studies [39,40]. An increased filtration could also be related to the free albumin permeation through large pores of a diameter of 75–110 Å, the presence of which in the glomerular barrier is suggested by Ohlson et al. [32] in the study on protein filtration in an isolated perfused kidney. The occurrence of such pores in vivo was confirmed in studies on patients with Fanconi syndrome, in whom significant amounts of larger proteins, such as transferrin or IgG, were found in urine [30].

However, the results obtained by Russo et al. [33] were regarded as not reliable by the scientific community. Gekle [41] pointed out that while the two-photon microscopy technique can be successfully used to assess the filtration of low-molecular-weight compounds, in the case of albumin, it could lead to misleading observations. With such a large difference in the concentration of labeled albumin in the plasma and lumen of the tubule, the fluorescence signal from the filtrate might have been significantly increased by background noise. In addition, the author emphasized that the proximal tubule reabsorption system was probably not efficient enough to transport such a large amount of albumin. Moreover, de Borst [42] noted that the presented results do not make it possible to come to such an unambiguous conclusion and suggested that the observed effect could be triggered by the direct toxic action of large amounts of albumin on the proximal tubular epithelial cells. It has been shown that long-term albumin exposure reduces endocytosis in proximal tubule cells [43]. Moreover, other studies reported that high albumin levels induce apoptosis in proximal tubule cells through reduction of megalin, protein kinase B, and Bad protein [44]. In turn, in the commentary of Remuzzi et al. [45], the authors included the results of the micro-puncture on *Munich-Wistar* rats, which are consistent with the previously determined GSC values and which have never before been published. In the case of the aforementioned rats, a subset of glomeruli were located on the surface of the kidney, which allowed the extraction of ultrafiltrate directly from the Bowman’s space, thus excluding the effect of reabsorption in the first section of the tubule [46]. Certainly, further studies are needed to clearly define the real GSC values for albumin filtration.

### 3.2. Albumin Reabsorption

The site of albumin reabsorption is well recognized. The studies that have been carried out since the beginning of the 1960s, using many microscopic techniques, undoubtedly indicate that this process occurs in the proximal tubule, being most effective in its early segment [26]. By the use of in vivo micro-puncture, it was established that albumin uptake takes place in a similar amount in the early and late part of the proximal convoluted tubule, and also partially in the descending part of the straight tubule. No significant albumin absorption was observed in other sections of the nephron [27]. However, the mechanism of albumin uptake remained unexplained for a long time. The proximal tubule is lined with a dense cuboidal epithelium. Thus, the intracellular penetration of albumin through the epithelium into the bloodstream seemed unlikely from the beginning. It would also be hindered by its relatively large size and low concentration in the filtrate [46]. The general kinetic characteristics of the albumin absorption process were provided by pioneer studies of Park and Mack in 1984 [47] using isolated perfused renal tubules. They indicated the presence of two uptake systems: the first one, a high capacity-low affinity system, characterized by Michaelis constant (K_m_) 1.2 mg/mL and (B_max_) 3.7 ng/min per mm tubule length and the second one, a low capacity system-high affinity, with K_m_ of 0.031 mg/mL and binding capacity B_max_ of 0.064 ng/min per mm tubule length.

Studies on the endocytosis process in the fluid phase (pinocytosis) in proximal tubular epithelial cells showed that it continues with too little efficiency to explain the effective uptake of albumin. Absorption of this protein occurs almost 40-fold faster than in the case of compounds absorbed in the pinocytosis process, such as dextran or inulin [48].

Experiments conducted in the 1990s demonstrated that the uptake of albumin is largely a specific process. Binding of labeled albumin can be almost completely inhibited by the excess of unlabeled molecules. In addition, unlabeled albumin enhances the dissociation of labeled molecules from the proximal tubular epithelial cell membrane. They are characterized by the dissociation constant K_d_ in the range of 100–300·10^−9^ M (7–20 mg/L). The albumin binding kinetics indicates the presence of at least one binding site. This characteristic suggested that the main role in albumin reabsorption is played by adsorptive endocytosis. It was confirmed that pharmacological inhibition of this process in rats by alkalization of endosomes with NH_4_Cl or bafilomycin A1 results in increased albumin excretion in urine [43,49].

In addition, studies on proximal-tubule-derived opossum-kidney cells (OK cells) demonstrated that albumin endocytosis depends on cytoskeleton integrity. Inhibition of actin polymerization by cytochalasin D results in an almost complete arrest of albumin absorption. Apparently, the process is also accelerated by interactions with microtubules. A marked decrease in the absorption of albumin was observed as a result of microtubule disruption due to the action of nocodazole, but the cessation was not complete. It seems that the movement of vesicles from the cell membrane towards the endosomal compartment in the initial phase of endocytosis, depends on the integrity of the actin skeleton, and interactions with microtubules are observed in the later phase. On the other hand, it was demonstrated in these studies, that clathrin-dependent endocytosis is the predominant mechanism of molecules reabsorption [50,51]. Although caveolin expression was detected in the apical membrane of proximal tubular epithelial cells, caveolin dependent endocytosis was never reported [52,53].

Only recent studies have explained the molecular mechanism of albumin uptake in the proximal tubule to a great extent. It turns out that the interaction of cubilin with amnionless protein (AMN) is necessary for the efficient uptake of this protein in the proximal tubule. Due to the close association and functional dependence of this complex, it is referred to as CUBAM. In addition, a proper expression and efficient operation of this complex depends on the second endocytic receptor—megalin [54,55,56]. Normal renal expression and function of CUBAM and megalin are necessary for the efficient uptake of albumin in proximal tubular cells, as can be seen from evidence of albuminuria in several inherited or acquired diseases concerning these endocytic receptors (Table 1).

The overall structures and functional associations of the receptor complex are presented in Figure 1. Recently, it was shown in studies using hepatocyte nuclear factor 1α knockout mice, that this protein controls the constitutive expression of both receptors [63].

Megalin, cubilin and amnionless present known domains and motifs. Megalin binds a variety of filtered proteins through its complement type repeats and is able to internalize ligands via NPXY motifs in the cytoplasmic tail. Cubilin contains multiple binding domains (CUB domains) and epidermal growth-factor-type (EGF-type) repeats, as a peripheral membrane protein depends on megalin and/or AMN. AMN contains an NPXY motif and probably assists cubilin in endocytosis as well as in intracellular transport during synthesis.

### 3.3. Cubilin

Cubilin was identified as the albumin receptor by Birn et al. in 2000 [64]. The authors isolated cubilin from the membranes of rat kidney brush-border using affinity chromatography with immobilized albumin. They also determined the dissociation constant of the complex by surface plasmon resonance (K_d_ = 0.63 μM) and demonstrated that inbred dogs with a functional defect of cubilin and mice with inhibited megalin expression excrete significant amounts of albumin in urine. Moreover, they also indicated the mutual interaction of both receptors in albumin endocytosis.

Cubilin is an extracellular glycoprotein with a molecular weight of about 460 kDa. Initially, it was identified as the antigen of teratogenic antibodies produced in rabbits after injection of renal brush border. Originally known as glycoprotein 280 (gp280), it was located in the proximal renal tubules and the yolk sac epithelium. Its expression was also found in the ileum, uterus and placenta. Initially, no ligands were attributed to this protein. Later studies showed that cubilin was identical to the receptor of the intrinsic factor-vitamin B_12_ (IF-B_12_) complex present in the small intestine. The function of a receptor protein was assigned to it [65]. The human cubilin gene was located on the 10p12.33-p13 chromosome. This protein is composed of a 110 amino acid N-terminal part, followed by 8 epidermal growth-factor-like (EGF-like) domains and 27 domains similar to complement C1r/C1s, epidermal growth-factor-related sea urchin protein (Uegf) and bone morphogenic protein 1 (CUB). Cubilin does not have a transmembrane domain. The N-terminus is responsible for protein embedding in the membrane. Each CUB domain consists of 110 amino acid residues. The structure of CUB domains consists of two layers of five antiparallel β-sheets also connected by β-structures. They constitute the most conservative area of the molecule and are responsible for ligand binding. In addition to albumin and IF-B_12_, iron-containing proteins (transferrin, hemoglobin, myoglobin), apolipoprotein A1 and vitamin D binding protein may also be bound [56,59,66,67]. Analysis of albumin binding to in vitro expressed cubilin fragments made it possible to identify the binding sites within CUB domains 6–8 [68]. These sites are likely to partially overlap with the binding sites of hemoglobin as demonstrated in a competition study using HK-2 cells (human kidney proximal tubule, adult male, papilloma immortalized) [69].

Cubilin is not a fully independent endocytic receptor. For its proper membrane localization and stability, it must interact with the AMN protein. In addition, this protein is responsible for the cubilin internalization. Recently, it was demonstrated that the FXNPXF sequence in the AMN cytoplasmic domain is functionally active and mediates sequestration of the complex by interacting with adaptor proteins: the disabled protein-2 (Dab2) and autosomal recessive cholesterolemia protein (ARH). CUBAM internalization may also occur through association with megalin [55,70]. Megalin seems to be responsible for cubilin maintaining in the cell membrane. Simultaneous shedding of cubilin from the membrane and its loss with urine were found in a patient with the Donnai-Barrow/FOAR syndrome (Facio-Oculo-Acustico-Renal Syndrome) due to a functional mutation in this receptor gene leading to its cytoplasmic retention [71].

The main role of CUBAM in tubular albumin uptake was demonstrated in recent studies in mice with renal-specific knockout of cubilin and megalin, in which moderate tubular albuminuria was observed [54,72]. This observation was also confirmed by the genetic and phenotypic analysis of patients with Imerslund-Gräsbeck syndrome, caused by a mutation in the cubilin or AMN gene. In both cases, albumin and other ligands of cubilin were lost in the urine [58].

Interestingly, cubilin and megelin have been also detected in rat and human glomeruli according to a pattern similar to that reported for megalin. Cubilin revealed both surface and intracellular expression in the podocytes [73]. Moreover, cubilin mRNA expression was detected in mesangial cells, but the absence of amnionless implied that cubilin is not fully functional in these cells. Nevertheless, albumin endocytosis was significantly impaired under inducible megalin knockdown conditions in stably transduced mesangial cells [74]. These findings may be important for better understanding of glomerulopathies.

### 3.4. Megalin

Megalin was identified as the albumin receptor by Cui et al. in 1996 [75]. The study was performed in vivo by injecting proximal tubules of rats with albumin solutions labeled with colloidal gold or ^125^I iodine isotope. In the experiment, the influence of many compounds on the tubular uptake of these proteins was determined. Administration of receptor-associated protein (RAP), cytochrome C reduced and gentamicin completely inhibited tubular reabsorption of labeled albumin. These ligands are characterized by high affinity for megalin, and their presence effectively hampers the binding of albumin through this receptor. The same applied to EDTA, a chelator of Ca^2+^ ions which also inhibited the uptake. This indicated that megalin contributes to the tubular reabsorption of this protein. These conclusions were confirmed by affinity chromatography on a column with immobilized megalin.

Megalin was originally discovered as an antigen in Heymann-type membranous nephropathy and called glycoprotein 330 (gp 330). In 1994, after a successful cloning attempt, its structure was better known, and the name was changed to the currently used one [76]. The human megalin gene was located on the chromosome 2q24-q31. Megalin is a large transmembrane protein which belongs to the family of low-density lipoprotein receptors. In proximal tubules, megalin is found in the apical membrane and clathrin-coated endocytic vesicles. This protein has a molecular weight of 600 kDa and is made up of about 4600 amino acid residues. The megalin molecule contains a large N-terminal extracellular domain, a single transmembrane domain and a short C-terminal cytoplasmic fragment. The extracellular domain has four areas rich in cysteine and is responsible for ligands binding. Several repeats of sequence similar to epidermal growth factor (EGF) and a cysteine-poor fragment containing YWTD motif separate the ligand binding region from the embedding portion. The signaling sequence is responsible for the dissociation of the ligand in the acidic endosome environment. The cytoplasmic part contains three NPXY motifs that initiate the process of albumin endocytosis by concentrating receptors in coated pits via the disabled protein-2 (Dab2) adapter protein [77]. Recruitment of this adapter is in turn dependent on phosphorylation catalyzed by protein kinases B (Akt1 and Akt2) [78,79].

Megalin also has the ability to bind many other proteins. Besides albumin, its ligands include inter alia: apolipoproteins (B, E, HJ), vitamin transporting proteins (A, D, B_12_), hormones (angiotensin II, epithelial growth factor, leptin), enzymes and their inhibitors (lysozyme, plasminogen) and others [80]. In the case of albumin, the binding sites on the receptor molecule have not been mapped yet. It also internalizes the CUBAM complex and its ligands, including albumin. The complexes formed are internalized with the formation of early endosomes. After acidification of the endosome environment, the complex dissociates and a free receptor is recycled to the cell surface via dense apical tubules [59,66,67,81].

It is worthy mention that high glucose regulates megalin-mediated albumin endocytosis in renal proximal tubule cells. Elevated levels of plasma glucose inhibited megalin expression and consequently albumin endocytosis [61].

### 3.5. Neonatal Fc Receptor of Immunoglobulin Fragment

Recent studies on renal albumin absorption indicate the important role of neonatal Fc receptor (FcRn) in the process of albumin recovering from the filtrate and protecting this protein against intracellular degradation. FcRn is a heterodimeric transmembrane protein similar in its structure to major histocompatibility complex (MHC) class I molecules. It is common in endothelial and epithelial cells. Its presence was confirmed in glomerular podocytes and in proximal tubules. It has the ability to independently bind albumin and immunoglobulins IgG in an acidic environment (pH 5–6). An increase in the pH value above 7 results in the release of ligands from the receptor [82]. FcRn in podocytes is present in vesicles belonging to the acidic endosomal compartment and is responsible for the binding of immunoglobulins and albumin that passed through the glomerular capillary endothelium to the filtration space. The filtered proteins are absorbed by the podocyte through endocytosis and/or pinocytosis, and then bound by FcRn at the acidic pH of the endosomes. Endocytic vesicles laden with proteins undergo transcytosis toward the lumen of the Bowman’s capsule. In conditions of increasing pH, the protein-receptor complex is dissociated, and the intact protein is released from the cell to the primary filtrate. This process is supposed to protect narrow filtration slits from clogging with large proteins [83,84]. Part of the albumin, filtered and reabsorbed in the proximal tubules, is not a subject of degradation in the lysosomes of the tubular epithelium and is most likely recovered in the transcytosis process. In this theoretical model, FcRn binds albumin, which is released from its receptors, presumably the megalin/cubilin complex in the acidic endosome environment, and then is directed through transcytosis to the basal site of tubular epithelial cells, where it is released into the interstitial tissue as a consequence of increasing pH [85]. Although, in the light of current research results, the recovery of albumin by transcytosis involving FcRn in proximal tubules takes place, this is a marginal phenomenon for normal albumin homeostasis in the body [56].

## 4. Intracellular Transport and Degradation of Albumin

### 4.1. Transport to Lysosomes and Degradation

The intracellular flow of albumin proceeds through vesicular transport. Thanks to the early experiments, it has been already discovered that the sorting of albumin, its receptors and the vesicles themselves is closely related to their environment acidification. Alkalization of vesicles with bafilomycin A1 or NH_4_Cl solution led to a rapid reduction in albumin absorption, even at low concentration. At the same time, both the breakdown of albumin in lysosomes and recycling of receptors to the cell membrane were slowed down [48,49,86].

Vesicles acidification is initially accompanied by clathrin coat dissociation and formation of early endosomes. Further acidification results in the dissociation of albumin from complexes with megalin and cubilin, and the budding of dense apical tubules that turn the receptors back to the cell surface. Dissociation of complexes is already observed below pH = 6.5 [26,75,81].

The transporters involved in the acidification of endocytic vesicles were identified in the recent years. V-ATPase located in the membrane of this organellum is involved in the process of protons transfer inside the endosome [87]. On the other hand, it is necessary to introduce negative ions to compensate for the potential of the endosomal membrane. In proximal tubule cells, chloride CLC-5 channels play an important role in the transport of the opposite ion. CLC-5 was demonstrated to be associated with albumin-containing endosomes. It cannot be found in dextran containing endosomes, although V-ATP-ase is present in both types of endosomes. Lowering the CLC-5 expression inhibits albumin endocytosis in the proximal tubule but does not affect the first phase of dextran endocytosis. The study carried out on mice with the CLC-5 gene knockout confirmed its importance in megalin and cubilin recirculation. The participation of CLC-5 in the endocytosis process is also of an interest for inherited renal diseases. Dent’s disease, in which one of the first symptoms is proteinuria, is caused by a mutation of CLC-5 coding gene [88,89].

The Na^+^/H^+^ antiporters (sodium/proton exchangers—NHEs), which remove H^+^ ions from the inside of the cell with simultaneous Na^+^ ion uptake, are also involved in the acidification of the endosomal vesicle environment. They are transporting molecules found in the plasma membrane of many cells, and their task is to maintain the correct pH inside the cell. These proteins consist of two functional domains, namely a hydrophobic domain containing several transmembrane sequences and hydrophilic domain responsible for the regulation of their transport activity [90]. The NHE-3 protein is expressed in intestinal epithelial cells and in proximal renal tubules, always on the apical side. It is able to circulate between the cell membrane and the early endosomal compartment, contributing to the acidification of the vesicle contents. Internalization of the NHE-3 transporter occurs via clathrin vesicles [91]. NHE-3 is probably involved in the absorption of Na^+^ ions in these cells. Endosomal NHE-3s play a major role in the process of albumin molecules endocytosis. Na+ gradient across the endosomal environment is supposed to dissipate along the endosomal pathway, and thus NHE3 should be important for the early steps of endocytosis, possibly prior to the stage(s) where H+-ATPase exerts its function [91]. NHE-3 inhibitors cause disturbances in the acidification of early vesicles, which considerably delays the process of albumin endocytosis. These disorders are partially explained by the slowed return of receptors to the cell membrane due to disturbances in the process of their dissociation. Another reason may be the disruption of the formation and fusion of transporting vesicles through alkalization. In addition, NHE-3 was found to be responsible only for lowering the pH in the very early phase of endocytosis. Its pharmacological inhibition leads to a significant reduction in the efficiency of this process [92]. Additionally, obesity may downregulate the NHE-3 and Na^+^/K^+^ ATPase expression due lowered levels of p-Akt factor and consequently reduce albumin reabsorption and endocytosis [93]. Moreover, significant pool of NHE-3 exists as a multimeric complex with megalin in the brush border of the proximal tubule [94].

There are several mechanisms of endocytosis regulation. However, in the case of albumin reabsorption, they are poorly understood.

Calcium ions do not directly affect this process. Change in the cytoplasmic concentration of Ca^2+^ ions only slightly affects albumin uptake. However, their complete removal from the extracellular space results in a marked reduction in the ability of protein uptake (K_m_ increases from 0.1·10^−6^ to 2.5·10^−6^ M), without an effect on the maximum rate of absorption. The presence of calcium ions is likely to determine the effectiveness of the first stage of endocytosis, i.e., the ligands binding by receptors [95]. It is known that stimulation of protein kinase A (PKA) by cyclic adenosine monophosphate (cAMP), forskalin or parathyroid hormone results in a reduction of the total albumin reuptake capability but does not change the protein’s affinity for the receptor. This underlying mechanism related to the alkalization of early endosomes depends on the cAMP presence. The effect of phosphatidylinositol 3-kinase (PI-3K) on albumin endocytosis was also reported [92,96]. An inhibition of its activity through the use of wortmannin significantly weakens this process efficiency. An activity of phosphatidylinositol 3-kinase is related to the phase of ligands internalization. Proteins G are also involved in the signaling cascade associated with the albumin internalization. The uptake of albumin is increased in the epithelial opossum kidney cells (OK cells) transfected with the gene of the G _αi-3_ subunit. This effect was suppressed by the pertussis toxin as the concentration and incubation time increased. Thus, the role of G proteins seems to be closely related to the regulation of vesicular transport processes [56,97,98].

### 4.2. Lysosomal Degradation

Lysosomal hydrolysis of proteins takes place with the participation of a set of lysosomal proteases, collectively referred to as cathepsins. The most important role in proteins degradation is attributed to cathepsins B, H and L. These hydrolases are transported in the form of proenzymes to early endocytic vesicles involving the mannose-6-phosphate receptor and are activated as they maturate. Partial hydrolysis of proteins occurs already in endosomes, however, massive degradation takes place only in lysosomes. Immunocytochemistry studies show that at least several different populations of endosomes and lysosomes, in terms of cathepsin composition, can be distinguished in the epithelial cells of the proximal tubule [99]. Recent studies indicate that the fusion of albumin-containing endosomes with lysosomes depends on the membrane receptors of these organelles. Proteinuria occurs in mice with lysosomal SCARB2/Limp-2 knockout, belonging to the family of B-class scavenging receptors, despite the correct expression and internalization of albumin receptors. It was found in these mice that fluorescently labeled albumin was effectively absorbed after intravenous administration but did not collocate with lysosomal cathepsin B and was not degraded. A similar effect was also observed in wild-type mice with massive albuminuria induced by i.p. albumin administration, indicating that, at the level of the proximal tubule, the limited capacity of lysosomal degradation plays a more important role in the development of albuminuria than the ability of its reabsorption [100,101]. However, the study of Nielsen et al. [102] revealed that the lysosomal capacity is increased under proteinuria in a murine model of focal segmental glomerulosclerosis (FSGS), supporting previous findings that lysosomal digestion capacity can be increased under proteinuria [99].

Much of the experimental data suggest that proteins—including albumin—are completely degraded in lysosomes to free amino acids, which are then transported through the lysosomal and basolateral membrane to the bloodstream. In experiments using isolated, perfused proximal tubules of rabbit and ^125^I labeled proteins, almost all perfusate radioactivity is associated with their catabolite ^125^I-monotyrosine. Similar results were obtained when measuring the efflux of radioactivity from kidney sections perfused with radiolabeled protein preparations [26,47].

However, the results of studies in this respect are not consistent. Some authors suggest that lysosomal degradation of albumin is not complete, and its products are predominantly polypeptide chains of varying length, which are returned to the lumen of the tubule and excreted in urine. Osicka et al. [103,104] demonstrated that significant amounts of peptides derived from degraded albumin are found in urine after intravenous administration of ^3^H-albumin to rats. However, they did not specify the size of the observed fragments in their study.

Similar results were obtained by Gudehithlu et al. [105] by intravenous administration of ^125^I-labeled albumin to rats. Urine proteins were precipitated with trichloroacetic acid and subjected to gel filtration allowing the determination of both fragments and whole albumin molecules. Small amounts of native albumin (2%) and significant amounts of its fragments (98%) with a molecular weight of 5–14 kDa were found in urine. These results also confirmed in vitro experiments on human HK-2 proximal tubule cells and ex vivo using perfused rat kidney. The amount of albumin excreted in the form of whole molecules and polypeptides was about 70-fold higher than the values determined so far. Low-molecular fragments of albumin in urine were also found in the studies in rats with diabetes caused by streptozotocin administration [35].

These discrepancies may be related to the use of radioimmunological methods in previous studies that do not detect polypeptide fragments lacking specific epitopes to the applied antibodies, which could happen in case of albumin degradation products. It is worth noting that such amounts of albumin excreted in urine would correspond to the amounts of filtered albumin with the assumption of a relatively large GSC, as discussed above.

Recent studies suggest that the albumin fragments observed in urine are not renal catabolites, but are the effect of free glomerular filtration, escaping the tubular resorption in a way not recognized until now. Both in control mice as well as in mice with cubilin or megalin knockout, mainly low-molecular albumin metabolites were found in urine after ^125^I albumin administration. The inhibition of tubular endocytosis did not affect the form of albumin present in urine. Generation of low-molecular albumin fragments was also observed in the plasma [106].

### 4.3. Transcytosis

The role of transcytosis as an alternative way of the internalized albumin targeting was proposed by Russo et al. in 2009 [33]. In their study, using electron microscopy and colloidal gold-labeled albumin, the authors observed the presence of albumin not only in lysosomes but also in large vesicles with a diameter of around 500 nm. These structures occurred throughout the cell and were often fused to the invagination of the basolateral membrane, releasing its contents into the peri-tubular network of capillary vessels. The integrity of the molecules was checked by means of antibodies directed against various albumin fragments. The authors suggest the co-existence of two mechanisms of intracellular albumin sorting depending on the integrity of molecules. Intact protein molecules return to circulation through a fast and efficient transcytosis, while a small part of molecules exhibiting structural changes are transported to lysosomes, where, according to the classic model, they are broken down into amino acid residues and in this form they return to the circulation [33,38]. The model aroused numerous controversies. Opponents of this theory, taking into account many years of microscopic examination of proximal tubule cells, undermine the existence of these types of vesicles and the possibility of their fusion with the basolateral membrane. They suggest that the structures observed are the result of artifacts related to tissue fixation by immersion. Methods based on perfusion, considered to be optimal, have been used so far [41,107]. On the other hand, recent studies on nephrotic mice with podocin, megalin and cubilin knockout indicate that endocytosis, dependent on these receptors, is not significant for maintaining albumin homeostasis. Despite total inhibition of reabsorption depending on these receptors, plasma albumin levels remained the same [108].

## 5. Conclusions

The kidney is responsible for about 10% of the albumin pool catabolized daily. This fraction is significant for homeostasis. Possible kidney function abnormalities, initially caused by ongoing damage to glomeruli, can eventually lead to severe failure of this organ and consequently to irreversible tubular damage. The secondary toxicity is attributed to dysregulation of epithelial cell functions due to protein overload under conditions of excessive endocytosis in proximal tubules [10]. Renal catabolism of proteins includes filtration in the glomerulus, adsorption and endocytosis in proximal tubules and intracellular hydrolysis or transcytosis of absorbed molecules.

The commonly accepted model assumes a relatively low level of glomerular albumin filtration due to its relatively large size and low isoelectric point, efficient reabsorption by endocytosis, involving megalin and cubilin receptors, vesicular transport to lysosomes and hydrolysis to single amino acid residues. In this model, only a negligible amount of albumin is excreted in urine, and degradation products mostly return to circulation.

The studies that have been carried out over the last decade yielded many unexpected observations that can dramatically change views on the topic of renal albumin catabolism. Alternative mechanisms that affect both the quantitative and qualitative image of this process emerge from them. New studies indicate, inter alia, that albumin permeation can occur with 50-fold higher yield than suggested by previous studies. The proposed value of the GSC would then be 0.04, not below 0.001 like in the currently accepted model [33]. Other controversies concern the degree of lysosomal degradation of albumin and direction of metabolites excretion. Some observations support the view that lysosomal proteolysis of albumin mainly produces polypeptides with a molecular weight of 5–14 kDa, which are largely recycled to the lumen of the tubule and excreted in urine. Reports about the phenomenon of albumin transcytosis are also surprising. This mechanism would explain the efficient absorption and transport of larger amounts of filtered albumin. At the same time, these reports suggest that proximal tubule dysfunctions mainly contribute to albuminuria in the course of various diseases, and not, as previously thought, glomerulopathies. The current concepts and new findings concerning renal albumin handling are summarized in Figure 2.

New reports have sparked a heated discussion in the world of science. If confirmed in further studies, they may change the views on the mechanism of albumin homeostasis and the pathomechanisms of diseases associated with albuminuria, such as nephrotic syndrome, renal failure against hypertension, diabetic nephropathy and cardiovascular diseases.

It is worth mentioning that a variety of data in the field of proximal tubular reabsorption, lysosomal metabolism and intracellular processing have been collected during basic studies on the structure of kidney epithelial cells. The achievements have been summarized in several excellent reviews [109,110,111,112].

## 6. Perspectives

Taking into account the available data, we believe that alternative models are not well documented so far and development of new therapeutic strategies should be based on the currently accepted concept.

In order to verify the significance of currently accepted and alternative concepts further investigations are needed. Development of new animal models and more detailed clinical studies, involving relevant molecules, such as glomerular proteins, endocytic receptors, vesicular transporters and other endosomal/lysosomal proteins are warranted.

## Figures and Tables

**Figure 1 ijms-22-05809-f001:**
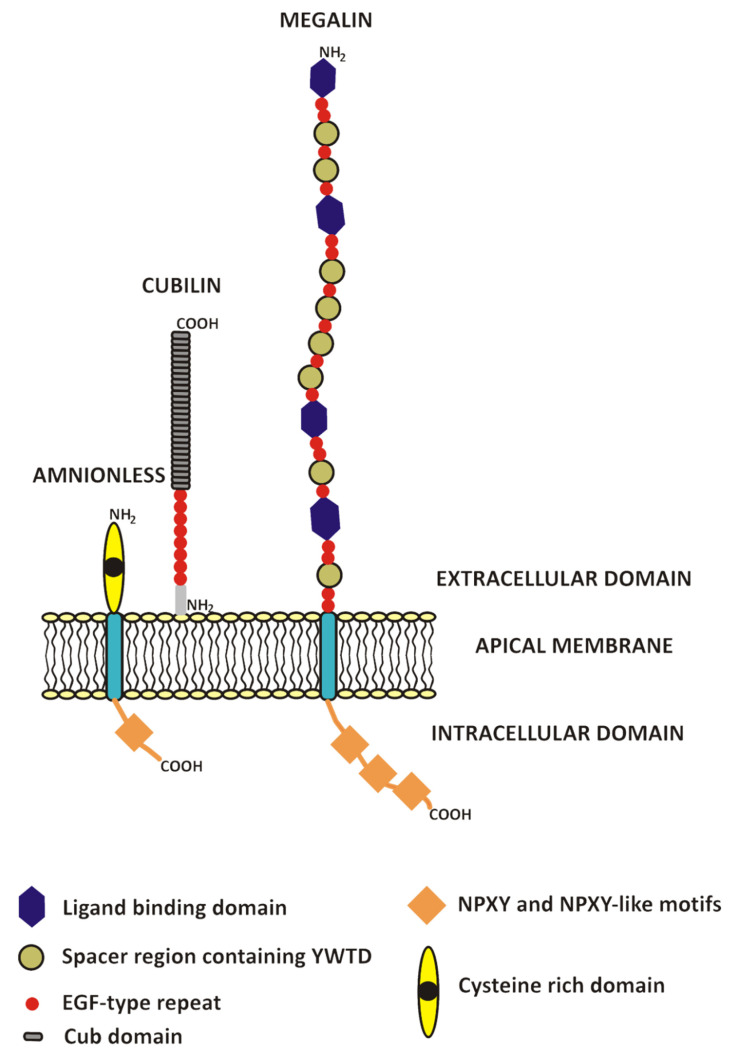
Endocytic megalin-cubilin-amnionless complex in the apical membrane of the renal proximal tubule.

**Figure 2 ijms-22-05809-f002:**
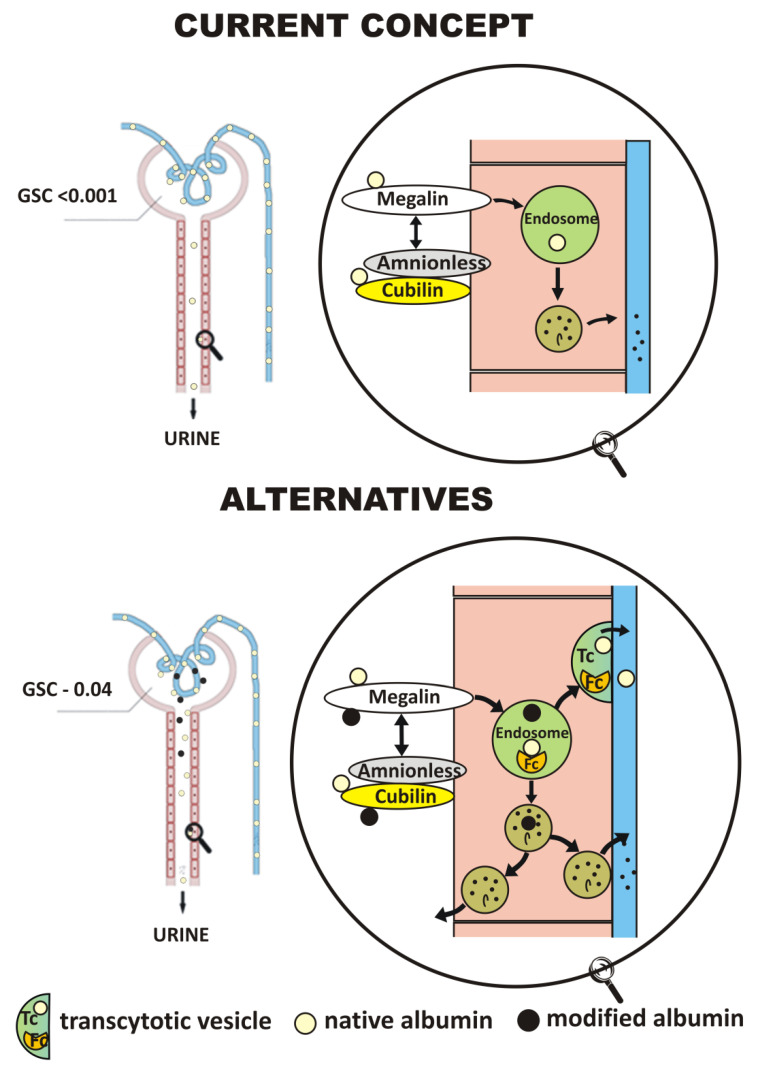
Renal catabolism of albumin—current concepts and alternatives. The generally accepted model assumes a relatively low level of glomerular albumin filtration due to its relatively large size and low isoelectric point, efficient reabsorption by endocytosis, involving megalin and cubilin-amnionless complex, vesicular transport to lysosomes and hydrolysis to single amino acid residues. In this model, only a negligible amount of albumin is excreted in urine, and the degradation products mostly return to circulation. The controversies concern both the quantitative and qualitative aspects of this process. New studies indicate, inter alia, that albumin permeation can occur with a GSC of 0.04. In the proposed model, native molecules undergo mainly transcytosis, while structurally modified molecules, i.e., oxidized or glycated, follow the classical path, with lysosomal degradation taking place to a limited extent. Most degradation products are short polypeptides that are secreted into the lumen of the tubule.

**Table 1 ijms-22-05809-t001:** Disorders associated with impaired renal handling of albumin.

Type of Disorders	Mechanism Implicated or Defective	Reference
**Donnai-Barrow facio-oculo-acoustico-renal syndrome (BBS/FOAR)**	Mutations in the *LRP2* gene encoding megalin cause reduction in the apical endocytic apparatus.	[57]
**Imerslund–Gräsbeck syndrome (megaloblastic anemia 1)**	Mutations in either the *CUBN* or the *AMN* gene lead to a defective expression or apical targeting of cubilin.	[58]
**Dent’s disease**	Mutations of chloride channel ClC-5 impair epithelial trafficking of membrane receptors and their ligands.	[59]
**Anti-LRP2 nephropathy**	LRP2 (megalin) has been identified as the antigen recognized by autoantibodies causing interstitial disease with segmental membranous nephropathy.	[60]
**Diabetes**	Glucose reuptake by proximal tubule inhibits protein kinase B activity leading to a decrease in megalin expression.	[61]
**Hypertension**	Microalbuminuria results from reduced functionality of the retrieval and/or lysosomal degradative pathways in the proximal tubule.	[62]

## Data Availability

Data sharing not applicable.

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
