# Peer review of "Renal Handling of Albumin—From Early Findings to Current Concepts"

_ijms, 2021, doi:10.3390/ijms22115809_

Round 1

Reviewer 1 Report

The paper reviewed the last decades of research and discussed the molecular mechanisms of albumin catabolism. Significantly, the protein is internalized in the proximal tubules via clathrin-dependent endocytosis with the tandem of macromolecular scavenger receptors - megalin and cubilin. Then the albumin molecules undergo lysosomal degradation and/or transcytosis. Better understanding the featured characteristics of albumin catabolism would open up the possibility of developing effective therapeutic strategies to treat proteinuric renal failure. Thus, the topic is of significance.

It is a very nice review, well written, covering the many aspects of the chosen topic neatly. There are only minor suggestions to the authors that may improve the quality of their excellent work.

  1. Figure 2, for the benefit of the non-specialist reader, please, explain briefly modified albumin.
  2. Line 560-561, “The proposed value of the GSC would then be 0.04, not below 0.001 like in the currently accepted model”. Reference is needed.
  3. It is worthy of mention that high glucose regulates megalin-mediated albumin endocytosis in renal proximal tubule cells (J Biol Chem . 2018 Jul 20;293(29):11388-11400). (Clin Exp Pharmacol Physiol. 2015 Oct;42(10):1118-26).
  4. Lines 192-194, “The lack of reabsorption was associated with a pronounced decrease in the expression of megalin, ATPase and clathrin in the apical pole of the tubule ”[27]. Please double-check the reference number. It appeared no megalin, ATPase, and clathrin shown in the Reference.

Additionally, does ATPase refer to P-type ATPase or V-type ATPase?

The paper mentioned the V-type ATPase. Is P-type ATPase sodium-potassium ATPase also involved in this endocytosis related to albumina?

Sodium potassium ATPase and NHE3 are both implicated in endocytosis related to renal sodium handling (Mol Cell Biochem. 2012 Aug;367(1-2):175-83). Is there any relationship between renal sodium and albumin handling in terms of endocytosis? 

  1. An additional paragraph, “Perspective,” would be helpful for the general reader in terms of future research direction.
  2. Make sure if there is extra space. If yes, please remove extra space. For example:

Line 71, “to------accumulate”;

Line 271, “have------explained”;

Line 326, “was------observed”;

Line 340, “immobilized------megalin”;

Line 388, “transcytosis------ to the basal site”;

Line 555, “and------ degradation,” etc.

Author Response

Dear Professor,

Thank you for your comments and suggestions. You will find below our response. We hope in this form the manuscript will be suitable for publication.

Best regards,

Krzysztof Gołąb

Response to the comments of Review Report 1.

Ad 1. Page 16, line 652; added text: oxidized or glycated.

Ad 2. Page 15, line 633; added reference [33].

Ad 3. Page 11, line 422; text added: It is worthy mention that high glucose regulates megalin-mediated albumin en-docytosis in renal proximal tubule cells. Elevated levels of plasma glucose inhibited megalin expression and consequently albumin endocytosis [61].

Ad 4. Page 5, line 212; reference changed for [35].

“Russo LM, Sandoval RM, McKee M, Osicka TM, Collins AB, Brown D, Molitoris BA, Comper WD. The normal kidney filters nephrotic levels of albumin retrieved by proximal tubule cells: retrieval is disrupted in nephrotic states. Kidney Int 2007; 71: 504-513.”

Page 5, line 212; The kind of ATPase was clarified - “V-ATPase”

Is there any relationship between renal sodium and albumin in terms of endocytosis?

According to our knowledge there is not any evidence that such relation exists. The endocytosis of   Na+,K+-ATPase is regulated by phosphorylation of adapter protein 2 (AP-2) trough kinase PKC-ç [Chen, Z.; Krmar, R.T.; Dada, L.; Efendiev, R.; Leibiger, I.B.; Pedemonte, C.H.; Katz, A.I.; Sznajder, J.I.; Bertorello, A.M. Phosphorylation of adaptor protein-2 mu2 is essential for Na+,K+-ATPase endocytosis in response to either G protein-coupled receptor or reactive oxygen species. Am. J. Respir. Cell Mol. Biol. 2006, 35,127-132], whereas in albumin internalization PKC-α is a key player [Hryciw, D.H.; Pollock, C.A.; Poronnik, P. PKC-alpha-mediated remodeling of the actin cytoskeleton is involved in constitutive albumin uptake by proximal tubule cells. Am. J. Physiol. Renal. Physiol. 2005, 288, F1227-F1235].

Ad 5. Page 17, line 665; the paragraph “Perspectives” has been added.

Ad 6. The extra spaces have been removed in whole text.

Reviewer 2 Report

The article is well written and easy to read. It faces with what is still considered an unsolved question in the kidney pathology: the role of proximal tubular cells in determining albuminuria one of the most important hallmark of kidney disease and a contributing factor to the development and progression of glomerulosclerosis and tubulointerstitial fibrosis, which impair kidney function. Clarifying the role of the glomerulus and of the tubule in albumin handling is mandatory, this manuscript summarizes the current concepts and unresolved questions.

My major concern regards on how authors use references  to substantiate their sentences.

In general, I believe that each sentence should be acknowledged by a reference, this not only for transparency but also for permitting readers to look at the article they are interested to. In the manuscript, this does not happen.

For example, in the introduction there is a bulk of sentences (from line 28 to 51) without any acknowledgement,   if not at the end of paragraph where two references are reported. The same is true for the section “general albumin metabolism”: from line 88 to 108, only one reference is reported; or for the sub-section “albumin reabsorption” where in the paragraph from lines 229 to 244 there is no reference attesting what is saying.

Below other examples:

At page 4, paragraph lines 178-186, it is not clear which of the two references the reported experiments refer to.

The sentence (lines 195-196) acknowledges a reference that does not testify what is said.

In the subsequent paragraph starting with “ The authors of the study….” it is not clear which Authors they refer to, those of reference 27 or those of references 28, 29 and 30.

Page 10 lines 420-441: there is a bulk of sentences having only two references (refs 75 and 76) despite the numerous and important  concepts they contain.  

In conclusion, I suggest Authors to review their manuscript and to check the concern I have addressed.

Other comments

  1. a section should be dedicated on the recent findings on how glomerular cells are able to handle albumin.
  2. the sentence affirming that caveolin-dependent endocytosis is not observed in proximal tubular cells (pag 6 lines 269-270, is not quite true. If on one hand, studies assessing this are quite lacking, except that of Dickson et al. (Dickson LE, Wagner MC, Sandoval RM, Molitoris BA. The proximal tubule and albuminuria: really! J Am Soc Nephrol. 2014 Mar;25(3):443-53), which adfirms that caveolin-dependent endocytosis is present in tubular cells, on the other hand immunohistochemical studies demonstrated the presence of caveolin in tubular cells of renal transplant recipients (Arpali E, Sunnetcioglu E, Demir E, Saglam A, Ozluk Y, Velioglu A, Yelken B, Baydar DE, Turkmen A, Oguz FS. Significance of caveolin-1 immunohistochemical staining differences in biopsy samples from kidney recipients with BK virus viremia. Transpl Infect Dis. 2021 Mar 22:e13605) ;
  3. page 10 line 394: “Transport to lysosome and degradation”, is it a subheading?
  4. clearer figure 2– the icons for proteins / organelles are too small to see

Author Response

Dear Professor,

Thank you for your comments and suggestions. You will find below our response. We hope in this form the manuscript will be suitable for publication.

Best regards,

Krzysztof Gołąb

Response to the comments of Review Report 2.

Major comments

Ad 1. Several references have been added, for example:

Page 1, line 34: [1]

Page 2, line 48: [2]

Page 2, line 87: [10]

Page 3, line 98: [11]

Page 3, line 103: [4]

Page 3, line 131: [19]

Page 4, line 157: [23, 24, 25]

Page 5, line 205: [33]

Page 5, line 239: [43]

Page 5, line 241: [44]

Page 6, line294: [52,53]

Page 10, line 376: [73]

Page 11, line 380: [74]

Page 13, line487: [90]

Page 13, line 505: [93, 94]

Page 15, line 618: [10]

Page 15, line 633: [33]

Additionally, few references have been given in added Table 1: [57-62].

Ad 2. Page 4, lines 198-219 : the references in paragraph have been corrected.

Ad 3. Page 5, line 212: the paragraph “the authors of the study….” has been linked together with the previous paragraph, making clear what it refers to.

Ad 4. Page 12, line 481: the references have been added: [90-94].

Other comments

Ad 1. Page 10, line 374: the section concerning handling of albumin by glomerular cells has been added:

“Interestingly, cubilin and megelin have been also detected in rat and human glo-meruli according to a pattern similar to that reported for megalin. Cubilin revealed both surface and intracellular expression in the podocytes [73]. Moreover cubilin mRNA expression was detected in mesangial cells, but the absence of amnionless im-plied that cubilin is not fully functional in these cells. Nevertheless, albumin endocy-tosis was significantly impaired under inducible megalin knockdown conditions in stably transduced mesangial cells [74].These findings may be important for better un-derstanding of glomerulopathies”.

Ad. 2 Page 6, line 292: The section concerning caveolin has been changed:

“Although, caveolin expression was detected in the apical membrane of proximal tubu-lar epithelial cells, caveolin dependent endocytosis was never reported [52,53]. References [52-53] have been added.

Ad 3. The title “Transport to lysosomes and degradation’ is the subheading – it has been changed to italic.

Ad 4. Page 16: the icons have been enlarged.

Reviewer 3 Report

My comments are in the enclosed file

Author Response

Dear Professor,

Thank you for your comments and suggestions. You will find below our response. We hope in this form the manuscript will be suitable for publication.

Best regards,

Krzysztof Gołąb

Response to the comments of Review Report 3.

Major comments

Ad 1. The title has been changed: “Renal handling of albumin – from early findings to current concepts”.

Ad 2. In addition to “Conclusion” section a short section “Perspectives” has been added.

Ad 3. In some phrases the years of findings have been added. For example: page 15, line 591; page 10, line 327; page 11, line 384.

Ad 4. The short expert opinion is presented in the “Perspectives” section.

Ad 5. Page 4, lines 148-160: the paragraph has been introduced:

It is formed from three layers: an innermost fenestrated endothelial cell layer, glomerular basement membrane, and an outermost layer of podocytes with their in-terdigitating foot processes bridged by a slit diaphragm. The multilayer characteristics of the sieve cause the pore gradually decreasing in size. In recent years, thanks to the development of genetic animal models, it was possible to identify the main compo-nents responsible for this barrier integrity.The most important role is currently at-tributed to some structural components of the glomerular basement membrane (GBM) (e.g., collagen type IV and laminin β2) and podocyte glycocalyx proteins (e.g. podocin and nephrin) [21, 22]. However, the function of podocytes is not limited to sieving. It has been demonstrated that both human and animal podocytes endocytose albumin in vitro [23, 24]. Data also suggests that podocytes clear proteins from the GBM via transcytosis [25]. These studies suggest that, podocytes internalize albumin and other plasma proteins (e.g. IgG) and in this way prevent the GFB from clogging.

Ad 6. Page 7: the Table 1 has been added.

Ad 7. Page  5, line 238: the sentence has been added: It has been shown that long-term albumin exposure reduces endocytosis in proximal tubule cells [43]. Moreover, other study reported that high albumin levels induce apoptosis in proximal tubule cells through reduction of megalin, protein kinase B, and Bad protein [44].

Ad 8. Some language improvements have been made.

Minor comments

Ad 1. Page 2, line 48: the paragraph has been modified as suggested:

“It is also worth mentioning that during long-term starvation albumin is degraded to provide essential amino acids for macromolecules synthesis and energy production”.

The role of cytokines and hormones in albumin homeostasis was detailed in sections Synthesis and Catabolism.

Ad 2. Page 2, line 56. Text was changed: “at pancreatitis, reduced synthesis in liver cirhosis”

Ad. 3. Page 2, line 62. We decided to generally describe the clinical manifestation of hypoalbuminemia. “hypotransmission” was changed to “hypotransferrinemia”.

Ad 4. Page 2, line 63: The sentence was modified according to suggestion:

“In addition, in the hypoalbuminemic states, the free, pharmacologically active fraction of drugs bound by albumin is increased, which should be considered for the optimal dosage”.

Ad 5. Page 2, line 70: The text was changed: “temporary correction of the oncotic pressure and prevents hypovolemia”.

Ad 6. Page 2, line 82: the text was added:

“The relation between proteinuria and renal failure is complex, and involves series of pathological events mediating by chemokines for interstitial inflammation, mononuclear cell accumulation, and intrarenal activation of complement. Proinflammatory and profibrogenic signals lead to local injuries and related interstitial renal fibrosis. As a consequence, the depletion of functional nephrons is observed [10]”.

Ad 7. Page 3, line 105. The paragraph has been modified:

“The synthesis of albumin is a continuous process which is regulated at the level of transcription and initiation of translation by different stimuli. For example, the syn-thesis is intensified after food intake and decreases in inter-meal periods. This process is also affected by hormones. The synthesis of albumin increases in hyperthyroidism and decreases in hypothyroidism. Corticosteroids and insulin enhance the production of albumin in healthy people.

Ad 8. Page3,  line 108:  the sentence has been modified: “ While the synthesis of this protein is inhibited in acute phase response”.

Ad 9. Page 4, line 164: changed as suggested to “controversies”.

Ad 10. Page 4, line 175: Changed as suggested to “observations in patients with rare diseases affecting albumin renal handling”.

Ad 11. Page 5, line 207: “xenobiotic” changed to “drug”.

Ad 12. Page 5, line 238: explained:

“It has been shown that long-term albumin exposure reduces endocytosis in proximal tubule cells [43]. Moreover, other study reported that high albumin levels induce apoptosis in proximal tubule cells through reduction of megalin, protein kinase B, and Bad protein [44].”

Ad 13. Page 6, line 263: the sentence was changed:

“They indicated the presence of two uptake systems: the first one, a high capacity-low affinity system, characterized by Michaelis constant (Km) 1.2 mg/mL and (Bmax) 3.7 ng/min per mm tubule length and the second one, a low capacity system-high affinity, with Km of 0.031 mg/ml and binding capacity Bmax of 0.064 ng/min per mm tubule length.”

Ad 14. Page 8, line 318: text added:

“Megalin, cubilin, and amnionless presenting known domains and motifs. Megalin binds a variety of filtered proteins through its complement type repeats and is able to internalize ligands via NPXY motifs in the cytoplasmic tail. Cubilin contains multiple binding domains (CUB domains) and epidermal growth-factor-type (EGF-type) repeats. As a peripheral membrane protein depends on megalin and/or AMN. AMN contains an NPXY motif and probably assists cubilin in endocytosis as well as in intracellular transport during synthesis”.

Ad 15. Page 10, line 350: changed to “iron-containing proteins”.

Ad 16. Page 11, line 395: changed to “membranous nephropathy”.

Ad 17. Page 15, line 614: the text was modified:

“Possible kidney function abnormalities, initially caused by ongoing damage to glomer-uli, can eventually lead to severe failure of this organ and consequently to irreversible tubular damage. The secondary toxicity is attributed to dysregulation of epithelial cell functions due to protein overload under conditions of excessive endocytosis in proximal tubules [10]”.

Round 2

Reviewer 2 Report

Authors have adequately addressed my concerns